# A Novel Interior Permanent Magnet Machine with Magnet Axis Shifted Effect for Electric Vehicle Applications

**Yongsheng Ge** , **Hui Yang \***, **Weijia Wang, Heyun Lin and Ya Li**

School of Electrical Engineering, Southeast University, Nanjing 210096, China; seu_emat_gys@163.com (Y.G.); seueelab_wwj@163.com (W.W.); hyling@seu.edu.cn (H.L.); seueelab_ly@163.com (Y.L.)
* Correspondence: huiyang@seu.edu.cn; Tel.: +86-15251867159

**Abstract:** This paper proposes a novel interior permanent magnet (IPM) machine with asymmetrical PM configuration. Different from the traditional IPM counterparts, the proposed machine can perform a magnet axis shifted (MAS) effect. The magnet axis is shifted towards the reluctance axis so that a higher resultant torque capability can be obtained. Firstly, the configuration and the basic principle of the proposed machine are described. The design parameters are optimized to improve the torque capability, and the effect of the PM asymmetry ratio on the torque performance is then evaluated in detail. In addition, the major electromagnetic characteristics of the optimized machine are investigated and compared with those of the Prius 2010 IPM machine by finite element method (FEM). The results demonstrate that the proposed asymmetrical PM configuration can achieve the torque improvement due to the MAS effect.

**Keywords:** asymmetrical permanent magnet configuration; interior permanent magnet (IPM); magnet axis shifted; permanent magnet (PM) machine





## 1. Introduction

Due to high torque density and high efficiency, permanent magnet (PM) machines have gained extensive attention recently. Amongst the conventional PM machines, interior PM (IPM) machines are regarded as popular choices for electric vehicles [1,2]. In order to improve the reluctance torque (RT) component, PM-assisted synchronous reluctance machines (PM-SynRMs) are extensively investigated in [3–5]. However, the conventional IPM machines suffer from a compromised utilization ratio of magnet torque (MT) and RT components, whose optimal current angles differ by 45 electrical degrees theoretically. In order to deal with this issue, dual rotor [6], hybrid rotor [7,8], and asymmetrical PM-SynRMs [4,9] were developed in recent years. However, the former two machines have relatively complicated structure and the latter one suffers from significant flux leakage as well as sophisticated magnetic flux paths.

The purpose of this paper is to propose a magnet axis shifted (MAS) IPM (MAS-IPM) machine with an asymmetrical PM arrangement, which aims to improve the torque component utilization ratio, and hence the torque density. In this paper, the configuration and the operating principle of the proposed machine are described. The effects of the PM asymmetry ratio on the torque performance are evaluated, and the design parameters are optimized to improve the torque capability. In order to validate the MAS effect of the proposed asymmetrical PM arrangement, the major electromagnetic characteristics of the optimal MAS-IPM machine are analyzed and compared with those of the Prius 2010 IPM machine (Toyota Motor Corporation, Tokyo, Japan) by finite element method (FEM).

## 2. Machine Configuration and Basic Principle

The configurations of the Prius 2010 IPM machine and the proposed MAS-IPM machine are shown in Figure 1a,b, respectively. For a fair comparison, the two machines share the same stator structure, active stack length, air gap length, current density and

PM usages, as shown in Table 1. The main feature of the MAS-IPM machine refers to an asymmetrical "▽"-shaped PM rotor structure, in which the lengths of the second-layer two PMs are unequal to achieve the MAS effect.

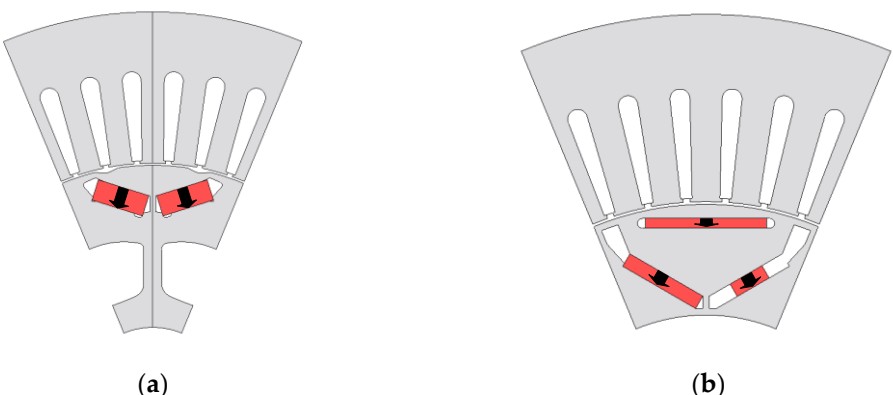

(**a**)  (**b**)

**Figure 1.** Machine configurations. (**a**) The Prius 2010 machine. (**b**) The MAS-IPM machine.

**Table 1.** Key common design parameters of Prius 2010 IPM and Mas-IPM machines.

| Items | Parameters |
|---|---|
| Stator outer diameter (mm) | 264 |
| Air gap length (mm) | 0.75 |
| Rotor outer diameter (mm) | 160.4 |
| Rotor inner diameter (mm) | 100 |
| Active stack length (mm) | 50.8 |
| Peak current (A) | 246 |
| Rated speed (rpm) | 3000 |
| PM volume per pole (mm$^3$) | 12,802 |

For the conventional IPM machine, the optimal current angles $\beta_{PM}$ and $\beta_R$ when the MT and RT components reach their peak values are different, the difference of which is expressed as

$$\gamma_s = \beta_{PM} - \beta_R \tag{1}$$

In the traditional IPM machine with a symmetrical rotor configuration, as we know, the current angle difference $\gamma_s$ is theoretically an electrical angle of 45. On the other hand, in the proposed machine, since the magnet axis is shifted towards the reluctance axis due to the asymmetrical PM arrangement, the difference of the current angles $\gamma_s$ can be significantly reduced, which can further improve the torque capability.

The no-load magnetic fields of the two machines for comparison are plotted in Figure 2. It can be seen that the magnet $d$-axis is shifted by an angle in the proposed MAS-IPM machine, while the reluctance $d$-axis remains unchanged. It means that the magnet and reluctance axes become closer compared with the Prius 2010 machine. As a result, the difference of the current angles $\gamma_s$ of the proposed machine can be reduced and the corresponding total torque is improved, confirming the feasibility of the MAS effect.

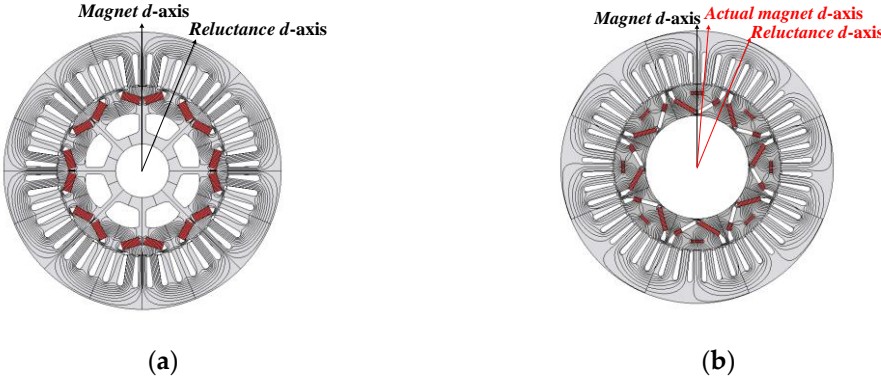

**Figure 2.** No-load magnetic fields. (**a**) The Prius 2010 machine. (**b**) The MAS-IPM machine.

## 3. Optimization of The Proposed MAS-IPM Machine

In order to maximize the torque capability by effectively employing the proposed MAS effect, some design variables are selected to be optimized, as shown in Figure 3. The peak current 246 A was used for obtaining the maximum torque, which is accordance with that of Prius 2010 machine. A coefficient $\alpha$ is defined as $l_{pm1}/l_{pm2}$ to describe the asymmetry level of PM1 and PM2, which is associated with the MAS effect. The design global optimization is performed by a multi-objective genetic algorithm with the constraints of the overall sizing, e.g., stator outer diameter and stack length, as shown in Table 1. The optimization target is to maximize the average torque and minimize the torque ripple, the corresponding weight factors of which are 1 and 0.5, respectively. In addition, the major design parameters of the optimized IPM machines and their variation ranges are presented in Table 2.

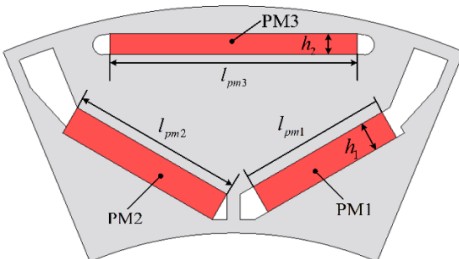

**Figure 3.** Design variables of the proposed machine.

**Table 2.** Definitions and variation range of the design parameters of proposed MAS-IPM machine.

| Items | Descriptions | The MAS-IPM Machine |
|---|---|---|
| $h_1$ (mm) | thickness of PM1 | 3.5~4.5 |
| $h_2$ (mm) | thickness of PM3 | 2.5~3.5 |
| $l_{pm1}$ (mm) | length of PM1 | 7~21 |
| $l_{pm2}$ (mm) | length of PM2 | 20~25 |
| $l_{pm3}$ (mm) | length of PM3 | 25~35 |
| $\alpha$ | $l_{pm1}/l_{pm2}$ (the asymmetry ratio) | 0.3~0.9 |

For illustrating the definition of the torque component utilization ratio, the diagram for the torque separation of a conventional IPM machine is shown in Figure 4. The torque component utilization ratios of the *MT* and *RT* $u_{pm}$ and $u_r$ are defined as follows

$$u_{pm} = \frac{MT_{comp}}{MT_{max}} \tag{2}$$

$$u_r = \frac{RT_{comp}}{RT_{max}} \tag{3}$$

where $MT_{comp}$ and $RT_{comp}$ are $MT$ and $RT$ when the total torque reaches the peak value, and $MT_{max}$ and $RT_{max}$ denote their maximum values of $MT$ and $RT$.

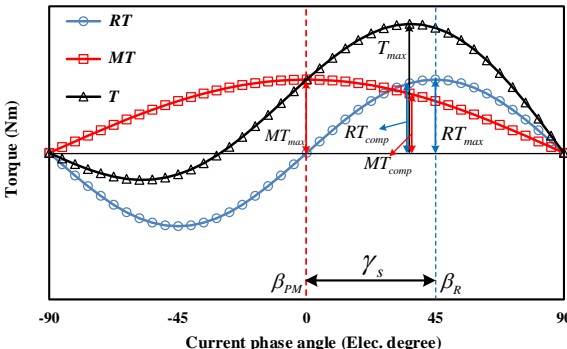

**Figure 4.** Diagram of torque component separation of a conventional IPM machine.

A preliminary model is obtained based on the global optimization. To better illustrate the influence of $\alpha$ on torque performance, the single parameter optimization is performed. The torque component utilization ratios of $MT$ and $RT$ as functions of different asymmetry coefficients $\alpha$ are presented in Figure 5. With the increment of $\alpha$, the $RT$ utilization ratio $u_r$ is increasing proportionally until $\alpha$ reaches 0.8, where $u_r$ reaches the peak value of 93.77%. At the same time, the $MT$ utilization ratio $u_{pm}$ decreases with the increase in $\alpha$, which is attributed to the weakening of the MAS effect in that case. Nevertheless, Figure 6 shows the toque component variation with the asymmetry coefficient. It shows that the total torque and the $MT$ component can be improved with the increment of $\alpha$ until it reaches 0.8. The improvement in the $MT$ component can be explained by the increase in PM usage, despite $u_{pm}$ decreasing. The $RT$ component is almost invariant because the rotor steel lamination remains unchanged.

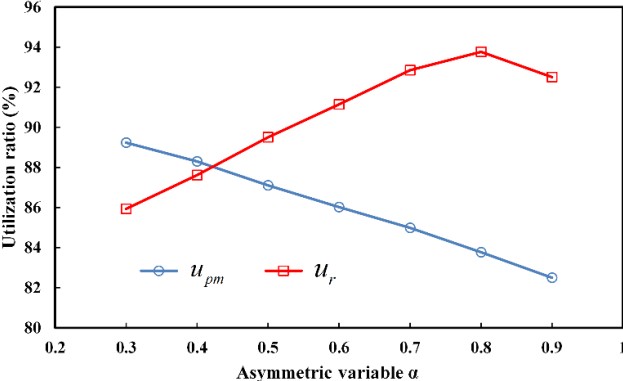

**Figure 5.** The torque utilization ratios versus the asymmetry coefficient $\alpha$.

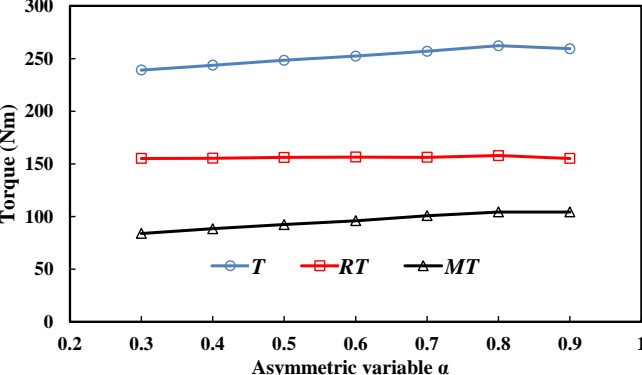

**Figure 6.** The total torque and torque components versus the asymmetry coefficient $\alpha$.

After the optimization, the optimal asymmetrical structure is confirmed. The optimized design parameters of the proposed MAS-IPM machine are obtained, i.e., $h_1$ = 3.9 mm, $h_2$ = 2.7 mm, $l_{pm1}$ = 18.4 mm, $l_{pm2}$ = 23 mm, $l_{pm3}$ = 33 mm, and $\alpha$ = 0.8.

## 4. Performance Comparison

### 4.1. No-Load Performance

The back-EMFs at rated speed of the two investigated machines are shown in Figure 7. It can be found that the root-mean-square (RMS) EMF of the proposed machine is 20.2% higher than that of the Prius machine. Meanwhile, it shows that the MAS-IPM machine exhibits better sinusoidal back-EMF than the conventional machine. The cogging torque waveforms are illustrated in Figure 8. It can be seen that the Prius 2010 machine exhibits higher cogging torque than the MAS-IPM machine. This is mainly attributed to lower THD of the back-EMF of the proposed MAS-IPM machine.

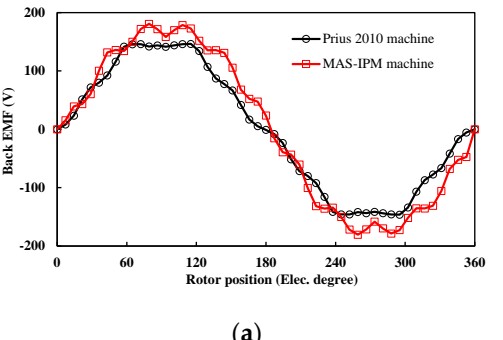　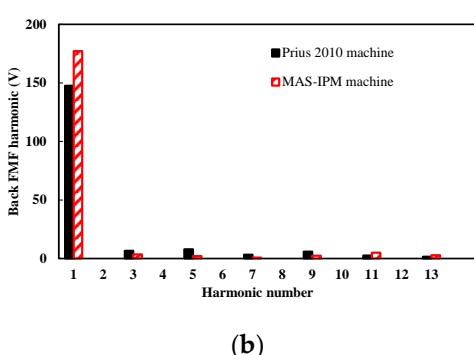

(**a**)　　　　　　　　　　　　　　　　　　　　　(**b**)

**Figure 7.** No-load back-EMFs of the two investigated machines (3000 rpm). (**a**) Waveforms. (**b**) Harmonic spectra.

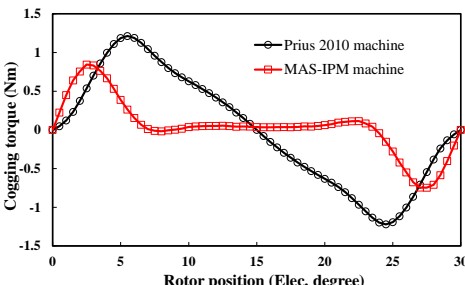

**Figure 8.** Cogging torque waveforms.

### 4.2. Torque Characteristics

The on-load torque characteristics of the Prius 2010 and the MAS-IPM machines are shown in Figures 9 and 10. The torque separation results obtained by the frozen permeability method [10,11] are shown in Figure 9a,b, respectively. Due to the asymmetrical PM configuration, $\gamma_s$ of the proposed MAS-IPM machine is reduced by 15 electrical degrees compared to the Prius IPM machine.

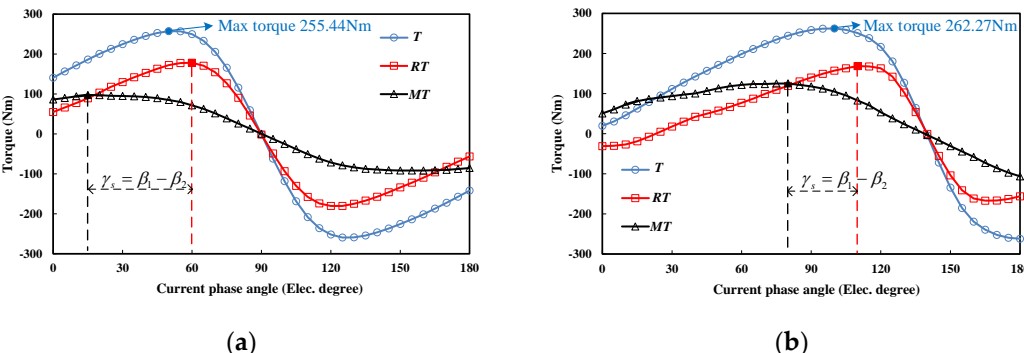

**Figure 9.** The torque component segregation. (**a**) The Prius 2010 IPM machine. (**b**) The MAS-IPM machine.

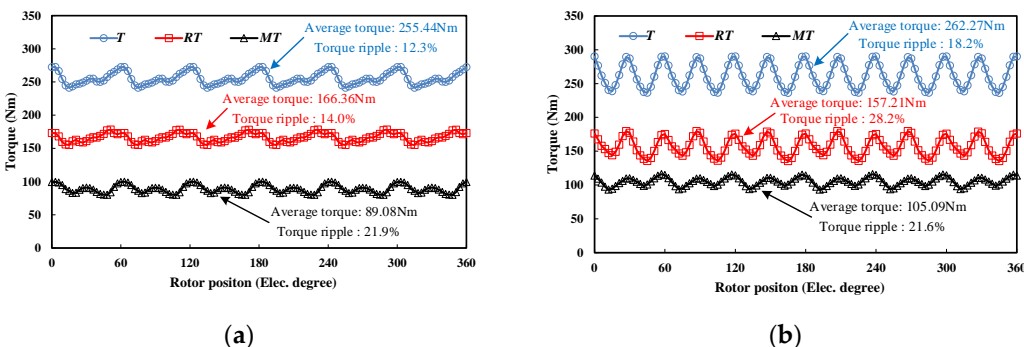

**Figure 10.** The steady-state total torque, reluctance torque and magnet torque under maximum torque conditions. (**a**) Prius 2010 machine. (**b**) MAS-IPM machine.

### 4.3. Torque/Power vs. Speed Curves

The field-weakening capability is a key characteristic for the traction machines for electric vehicles. The torque/power-speed curves are shown in Figure 11a,b, respectively. It demonstrates that the proposed machine can achieve higher torque and power capability over a whole speed range.

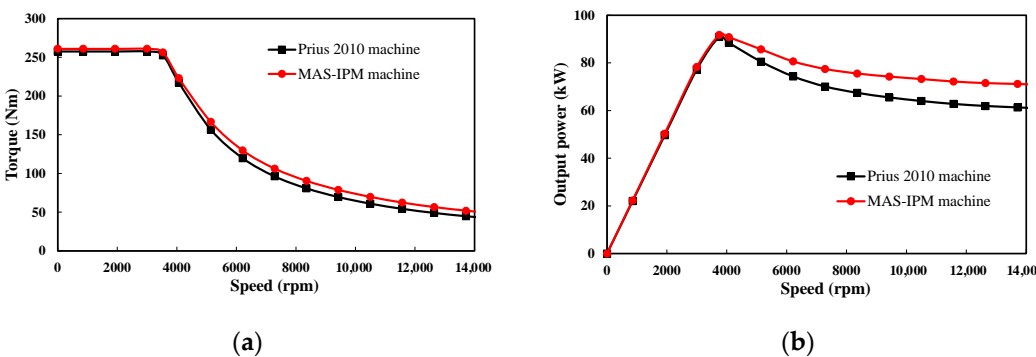

**Figure 11.** Torque/power-speed curves. (**a**) The torque-speed curve. (**b**) The output-speed curve. ($I_{max}$ = 246 A, $U_{dc}$ = 650 V).

### 4.4. Iron Loss and Efficiency Maps

The loss in the machine mainly contains the iron and copper losses. The iron, hysteresis, eddy current, and copper losses can be calculated by

$$P_i = P_h + P_e \tag{4}$$

$$P_h = K_h f B_m^{\alpha} \tag{5}$$

$$P_e = K_e f^\gamma B_m^\delta \tag{6}$$

$$P_c = 3R_a I_a^2 \tag{7}$$

where $P_i$, $P_h$, $P_e$, and $P_c$ are iron, hysteresis, eddy current, and copper losses, respectively; $K_h$ and $K_e$ are the coefficients of hysteresis and eddy current losses, respectively; $f$ is the operating frequency of the machine; $B_m$ is the flux density; $\alpha$, $\gamma$, and $\delta$ are the coefficients of the empiric formula; $R_a$ is the armature winding resistance; and $I_a$ is the phase current RMS value.

The iron losses of the two machines are given in Figure 12. It can be observed that the two machines show similar iron losses when the speed is lower than 6000 rpm. Besides, due to lower THD of the back-EMF, the MAS-IPM machine shows lower iron loss under high speed range. The efficiency maps of the two machines are illustrated in Figure 13. The maximum efficiency of the MAS-IPM machine is 0.7% higher than that of the Prius 2010. It can be seen that the high efficiency over a wider operating region can be achieved in the MAS-IPM machine due to its lower iron loss. The related electromagnetic performances of the two machines are given in Table 3. As a whole, the MAS effect of the proposed asymmetrical PM configuration for the torque improvement is confirmed. Meanwhile, compared to the Prius IPM machine, the proposed machine exhibits higher RMS flux linkage, RMS back-EMF and lower cogging torque, as well as higher MT, RT utilization ratio, and operating efficiency.

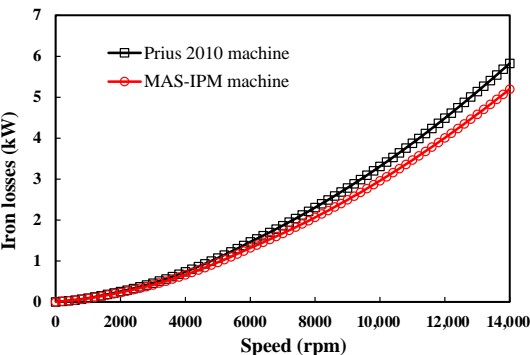

**Figure 12.** Iron losses versus speed curves.

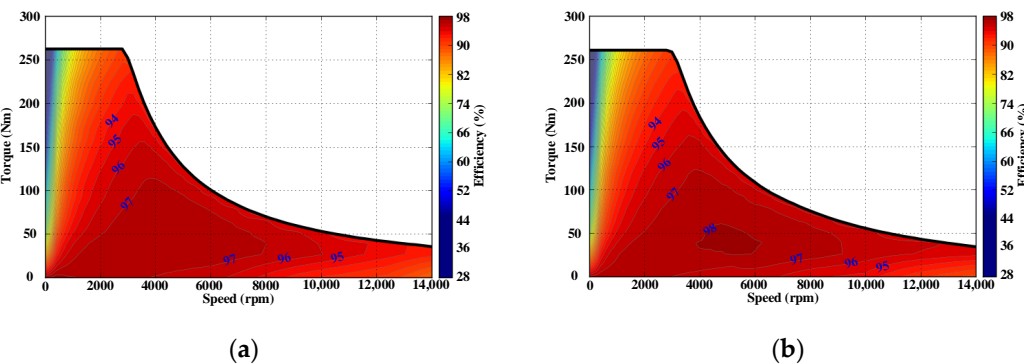

**Figure 13.** Efficiency maps. (**a**) The Prius 2010 machine. (**b**) The MAS-IPM machine. ($I_{max}$ = 246 A, $U_{dc}$ = 650 V).

**Table 3.** The electromagnetic performance of the two investigated machines.

| Items | Prius 2010 Machine | MAS-IPM Machine |
|---|---|---|
| RMS back-EMF (V) | 104.42 | 125.31 |
| Rated torque (Nm) | 255.44 | 262.27 |
| Reluctance torque (Nm) | 166.36 | 157.21 |
| Magnet torque (Nm) | 89.08 | 105.09 |
| Peak cogging torque (Nm) | 1.21 | 0.83 |
| $\gamma_s$ (elec. deg.) | 45 | 30 |
| Torque pulsation (%) | 12.3 | 18.2 |
| $u_{pm}$ (%) | 80.21 | 83.77 |
| $u_r$ (%) | 91.24 | 93.31 |
| Maximum efficiency (%) | 97.4 | 98.1 |

## 5. Conclusions

A novel IPM machine with a MAS effect is proposed in this paper. Due to the asymmetrical PM configuration, the proposed machine benefits from the reduced $\gamma_s$, improving the MT and RT utilization ratios. Hence, the total torque can be further improved. Based on FEM, the design variables of the proposed MAS-IPM machine are optimized to maximize the torque capability by defining an asymmetry ratio. Afterwards, the electromagnetic characteristics of the proposed MAS-IPM machine are investigated and compared with those of the Prius 2010 machine. It can be found that the proposed machine shows higher RMS back-EMF, lower total harmonic distortions, and lower cogging torque. In addition, the MAS-IPM machine exhibits a higher peak torque, a higher high-speed power maintaining capability, as well as wider high efficiency operating regions. In summary, the results confirm the MAS effect of the proposed asymmetrical PM configuration due to its performance improvement. A prototype of the MAS-IPM machine will be manufactured and the test results will be reported in due course.

**Author Contributions:** Conceptualization, Y.G., H.Y. and W.W.; methodology, H.Y., Y.G. and W.W.; software, H.Y., Y.G. and W.W.; validation, H.Y., Y.G. and W.W.; formal analysis, H.Y., Y.G. and W.W.; investigation, Y.L.; resources, H.Y., Y.G. and W.W.; data curation, W.W.; writing—original draft preparation, H.Y., Y.G. and W.W.; writing—review and editing, H.Y., W.W., and H.L.; visualization, H.Y., H.L. and Y.L.; supervision, H.Y., Y.G. and W.W.; project administration, H.Y., and H.L.; funding acquisition, H.Y. and H.L. All authors have read and agreed to the published version of the manuscript.

**Funding:** This work was jointly supported in part by National Natural Science Foundations of China under Grants (52077033 and 52037002), in part by Key R&D Program of Jiangsu Province (BE2021052), in part by the Fundamental Research Funds for the Central Universities (2242017K41003), in part by Supported by the "Zhishan Youth Scholar" Program of Southeast University, in part the Jiangsu Provincial Key Laboratory of Smart Grid Technology and Equipment, Southeast University (7716008046), and in part by supported by "the Excellence Project Funds of Southeast University".

**Conflicts of Interest:** The authors declare no conflict of interest.

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
