# Peer review of "A Novel Interior Permanent Magnet Machine with Magnet Axis Shifted Effect for Electric Vehicle Applications"

_wevj, doi:10.3390/wevj12040189_

Round 1

Reviewer 1 Report

  1. The article presents the optimization results, but there is no information on the method that was used . Which algorithm is used for optimization? How was the objective function defined?
  2. Are the geometric center of the PM1 and PM2 magnets always in the same position with the holes in the rotor? Can they move during optimization?
  3. How is the volume of the magnets in the Prius 2010 machine compared to the optimal version of the MAS-IPM machine?
  4. In Fig. 5, upm decreases with increasing α and in Fig. 6 MT increases. How should this be understood? Should one not flow from the other?
  5. 6 - for which current value were these results obtained?
  6. Line: 100-103- “The cogging torque waveforms are illustrated in Figure 8. It can be seen that the Prius 2010 machine exhibits lower cogging torque than the MAS-IPM machine.” The Prius has a higher cogging torque.
  7. Why is BEMF only shown at 1200 rpm? Isn't it better to present it for higher speeds?
  8. 10 - maybe it is worth optimizing machines also in terms of torque ripple?Because they have definitely increased.
  9. How was iron losses determined?
  10. It would be worth verifying the correctness of the FE models.

Author Response

Dear reviewer,

Yours sincerely

The authors

Reviewer 2 Report

No changes requested. The authors present a newasymmetric configuration for the magnets of an IPM synchronous machine, with the intent to improve the torque production. The configuration is assessed with FEM and compared with the machine used in Prius 2010. I think the paper can be considered suitable for publication.

Author Response

(The authors gave the same response as above.)

Reviewer 3 Report

Dear Authors,

The paper deals with an interesting topic and it is generally well written. I have two minor observations:

  • I do not see the relevance of the presentation of the volume of the permanent magnets.
  • The caption of Figure 9 is not correct: ‘The results of the torque separation.’ Please make the necessary correction.

Best regards

Author Response

(The authors gave the same response as above.)

Round 2

Reviewer 1 Report

Thanks for the answers